# Screening of Breast Cancer from Sweat Samples Analyzed by 2-Dimensional Gas Chromatography-Mass Spectrometry: A Preliminary Study

**DOI:** 10.3390/cancers15112939

**Published:** 2023-05-26

**Authors:** Michelle Leemans, Vincent Cuzuel, Pierre Bauër, Hind Baba Aissa, Gabriel Cournelle, Aurélien Baelde, Aurélie Thuleau, Guillaume Cognon, Nicolas Pouget, Eugénie Guillot, Isabelle Fromantin, Etienne Audureau

**Affiliations:** 1Clinical Epidemiology and Ageing Unit, Institut Mondor de Recherche Biomédicale, Paris-Est University, 94010 Créteil, France; etienne.audureau@aphp.fr; 2Forensic Institute of the French Gendarmerie, Caserne Lange, 5 Boulevard de l’Hautil, Cedex, 95001 Cergy-Pontoise, Franceguillaume.cognon@gendarmerie.interieur.gouv.fr (G.C.); 3Wound Care and Research Unit 26, Curie Institute, Rue d’Ulm, 75005 Paris, Francebabaaissa.hind@gmail.com (H.B.A.); aurelie.thuleau@curie.fr (A.T.); isabelle.fromantin@curie.fr (I.F.); 4Baelde & Cournelle Analytics, 130 Allée Reysa Bernson, 59800 Lille, France; gabriel.cournelle@gmail.com (G.C.); aurelien.baelde@gmail.com (A.B.); 5Department of Surgical Oncology, Curie Institute, 35 Rue Dailly, 92210 Saint-Cloud, France; nicolas.pouget@curie.fr (N.P.); eugenie.guillot@curie.fr (E.G.); 6Public Health Department, Henri-Mondor Hospital, Assistance Publique des Hôpitaux de Paris, 94010 Créteil, France

**Keywords:** chromatography, breast cancer, screening, sweat, mass spectrometry, machine learning

## Abstract

**Simple Summary:**

Breast cancer is the second most commonly diagnosed cancer among women. Novel breast cancer screening techniques are necessary to improve early detection and increase the chances of successful treatment. This study investigates the presence of specific volatile organic compounds in the sweat of breast cancer patients, which could serve as biomarkers for non-invasive screening. Sweat samples were collected from 21 breast cancer patients before and after tumor removal surgery and analyzed using thermal desorption coupled with gas chromatography and mass spectrometry. This work demonstrates that volatile organic compounds in sweat differ between pre-and post-surgery status in breast cancer patients, suggesting the possibility to use these compounds as novel breast cancer biomarkers and opening the way to new screening tools.

**Abstract:**

Breast cancer (BC) remains one of the most commonly diagnosed malignancies in women. There is increasing interest in the development of non-invasive screening methods. Volatile organic compounds (VOCs) emitted through the metabolism of cancer cells are possible novel cancer biomarkers. This study aims to identify the existence of BC-specific VOCs in the sweat of BC patients. Sweat samples from the breast and hand area were collected from 21 BC participants before and after breast tumor ablation. Thermal desorption coupled with two-dimensional gas chromatography and mass spectrometry was used to analyze VOCs. A total of 761 volatiles from a homemade human odor library were screened on each chromatogram. From those 761 VOCs, a minimum of 77 VOCs were detected within the BC samples. Principal component analysis showed that VOCs differ between the pre- and post-surgery status of the BC patients. The Tree-based Pipeline Optimization Tool identified logistic regression as the best-performing machine learning model. Logistic regression modeling identified VOCs that distinguish the pre-and post-surgery state in BC patients on both the breast and hand area with sensitivities close to 1. Further, Shapley additive explanations and the probe variable method identified the most important and pertinent VOCs distinguishing pre- and post-operative status which are mostly of distinct origin for the hand and breast region. Results suggest the possibility to identify endogenous metabolites linked to BC, hence proposing this innovative pipeline as a stepstone to discovering potential BC biomarkers. Large-scale studies in a multi-centered VOC analysis setting must be carried out to validate obtained findings.

## 1. Introduction

Breast cancer (BC) is the second most commonly diagnosed cancer among women. In 2020, the World Health Organization estimated that there were 2.26 million cases of BC worldwide [1]. Mammography is generally used for screening and studies affirm that the use of this technology can reduce mortality by up to 19% [2]. BC Screening is recommended for women between 50 and 69 years of age, with testing every two years. It should be noted that BC at a younger age is exhibiting an increasing incidence, and these cases are missed out in the current screening programs [3]. Additionally, screening programs may not be feasible in deficient health systems and resource-constrained settings. Therefore, an effort has been invested in discovering new diagnostic biomarkers to complement current screening programs. 

A promising avenue for the detection of disease biomarkers lies within volatile organic compounds (VOCs). The entire set of VOCs generated by an organism is called ‘volatilome’ and the study of the volatilome is known as “volatilomics”. Volatilomics emerged as a non-invasive approach, potentially low-cost, with a potentially larger coverage of the population for cancer screening [4,5,6]. To this end, the VOCs of interest result from metabolic activity and are found in patients’ biological samples, namely saliva, urine, breath, or sweat [4,7]. Distinct patterns of VOCs have been correlated with multiple diseases and cancers [8]. Therefore, the altered VOC pattern may serve as biomarkers for assessing or detecting the disease of interest [4]. 

Gas chromatography coupled with mass spectrometry (GC − MS) is the gold standard for identifying and quantifying human-expelled VOCs [4,9]. In GC − MS-based volatomics, VOCs are first separated on a GC column and then detected by MS. Comprehensive two-dimensional gas chromatography-mass spectrometry (GC × GC − MS) uses two GC columns, usually connected with a thermal modulator. The second column consists of a different stationary phase allowing for additional separation of VOCs that might coelute from the first column compared with classical GC − MS [10]. Nonetheless, GC × GC − MS has not been extensively employed for biomarker discovery, likely due to the difficulty of data analysis. 

This study aimed to establish the volatomic signature of BC. For this, we hypothesized that previously identified BC-related VOCs originate from the BC tumor [4] and can be detected in the sweat of BC patients. To this end, the sweat of BC patients before (so-called “sick status”) and after tumor-removing breast surgery (so-called “healthy status”) was analyzed with GC × GC − MS technology combined with multivariate statistical tools. By comparing the sweat volatome of BC patients in their sick states (before surgery, with tumors) with their healthy status (after surgery, without tumors), we aim to isolate specific VOCs that are indicative of BC. Identical patients before and after surgery were chosen to minimize the interference of epigenetic and external factors. Sweat samples were taken on the hand and breast area. First of all, we investigated if we could identify a BC-specific signature on the breast area and secondly, whether we could find a BC-specific volatomic pattern on the hands. The breast area was chosen based on its proximity to the breast tumor whereas the hand area was selected given both its easiness of sampling as well as for the advantage that hand VOCs have been previously thoroughly investigated and have served as the basis of the employed VOC database [11]. The utilized high-throughput VOC detection strategy might have the potential to be applied in a clinical environment as a complementary approach to the current diagnostic methods. 

## 2. Materials and Methods

The Ethics Committee ‘Comité de protection des personnes Ile de France III’ approved the study protocol (n° NCT04541537), and each patient signed an informed consent form before study enrollment.

### 2.1. Subjects & Study Design

BC patients were recruited during the first surgery consultation at Institute Curie (Paris, France). Inclusion criteria included: (1) patients that were seen in surgery consultation for an invasive non-metastatic BC treatment with breast-conserving surgery, with axillary or sentinel node dissection and (2) female patients over 18 years old, benefitting from social security. Exclusion criteria included: (1) wound presence on breasts and hands, (2) neoplasia in progress or neoplasia history of cancer other than breast, (3) pregnant or lactating women, (4) metastatic BC, or (5) concomitant medication taken one month before the surgical act (antibiotics, corticoids, anti-diabetics). A total of 21 women were included before surgical intervention and 13 women after surgery (see Table 1). The median age was 51 years for BC patients. Non-invasive odor sampling was conducted with an odor-sensing polymer (Sorbstar^®^) on the hands before and after surgical excision of the tumor. Other samples were taken on the diseased breast before and after the surgical ablation of the tumor. Each patient is its own control via inclusion before and after surgical excision, in an attempt to correctly isolate the BC chemical signature. The samples after surgery were made after healing and before the start of chemotherapy and/or radiotherapy treatment. No anti-cancer drugs were taken prior to sweat sampling. Only full-success tumor-removing surgery patients have been included in the study. No information on later BC re-occurrence has been collected. 

### 2.2. Sweat Sample Collection

For breast sweat collection, participants were given a kit containing a sterile cotton pad, a sterilized jar, unscented soap (Topialyse, SVR laboratory), a pack of sterile cotton tissues, and a tubular compression bandage. Each participant was requested to put the cotton pad in contact with the diseased breast after having showered with the provided soap. Drying of the breast was conducted with one of the given sterile tissues. On each cotton pad, four individual sorbents (SorbStars^®^ (Action Europe, Sausheim, France)) were stapled. This sorbent is a silicon-based polymeric phase and has the advantage of being stored in an inert atmosphere. From an analytical point of view, the polymer is much cleaner and provides less interference than a sterile cotton pad. A tubular compression bandage was requested to be worn on top of the cotton pad to keep the pad in place, with a minimum contact of 6 h between the diseased breast and the pad. Two samplings were conducted, one on the night prior to surgery and a second collection on the night before post-operative evaluation (median time after operation: 26 days). The cotton pad was removed and put into the sterile jar by the patient; patients washed their hands with the given soap prior to pad removal. The next day, participants brought the sample to the hospital. 

Hand sweat collection consisted of rubbing four individual SorbStars^®^ polymers for 15 min into the hands. To prevent contamination by exogenous VOCs, a protocol was implemented, consisting of prewashing the hands with perfume-free soap (Topialyse, SVR laboratory). After VOC collection, the participant was requested to put the sorbent within the glass collection vial. All hand sweat collections were conducted at the hospital and were collected one time before tumor-removing surgery and once after. 

### 2.3. Analytical Devices

After sample collection, the Sorbstars^®^ were thermodesorbed prior to GC × GC − MS analysis. For thermodesorption, the Versatile Sample Preparator (VSP4000) purge and trap system from Innovative Messtechnik GmbH (Vohenstrauß, Germany) was employed. The development and optimization of the analytical method were the topics of two previously published studies [11,12]. Volatile components were purged from the SorbStars^®^ by the carrier gas of the GC. The concentration step was completed by adsorption on a Tenax TA in the system trap by freezing out at −30 °C. After incubation of the sample at 190 °C and completing of the purging process (20 mL/min. for 20 min.), the concentrated substances were transferred by fast thermal desorption (TD) from the trap onto a transfer line, heated at 280 °C, and then separated by GC.

The thermodesorption device was coupled with a GC × GC − MS Q2010Plus purchased from Shimadzu. A DB-1MS column (Agilent, 30 m × 0.25 mm, 0.25 μm) coupled with a DB-1701 column (Agilent, 1.5 m × 0.1 mm, 0.1 μm) was used to conduct the chromatographic separation. The modulation was performed with an N2-cooled Zoex ZX1 thermal modulator, and the modulation time was set at 8 s. The initial temperature was set to 40 °C for 1 min, then raised to 250 °C at 2.5 °C/min, and held for 1 min at 250 °C. The MS was used with the electronic ionization source (70 eV) heated at 200 °C, the acquisition was performed in scan mode. The scan range was 29–250 *m*/*z*, and the sampling frequency was 50 Hz. No commercial standard solutions were used to confirm identification. To ensure the reliability of our analysis, several measures were taken. Firstly, blanks were included in every batch of analyses that were confirmed to be completely devoid of any screened compounds of the homemade library. Secondly, the consistency of the blank adsorbent phase (sorbstars) was thoroughly analyzed and confirmed. Furthermore, the potential issue of false positive peaks generated by the Sorbstars sampling medium was investigated. Our tests revealed that blank sorbstars exhibit visible silica peaks, which can also be detected in patients’ samples. However, the exclusion of these silica peaks from further processing was conducted without impacting the data thanks to the high peak capacity provided by the comprehensive gas chromatography. Additionally, each chromatogram was visually inspected to ensure that the cryogenic modulation was effective and that there were no anomalies present in the data. 

### 2.4. Data Analysis

#### 2.4.1. Extraction and Pre-Treatment of the GC × GC − MS Raw Data

Chromatogram processing was conducted as described by Cuzuel et al., 2018 [13]. In short, multiple individuals were asked to rub various sorbstars in their hands for 30 min, and the resulting samples were analyzed using GC × GC − MS. A total of 761 compounds were recorded in our so-called homemade library with their mass spectra. To assign formal names to the compounds, a comparison of the mass spectra with the NIST 14 library, with a minimum similarity threshold of 80%, was made. Please note that the research and the comparison within chromatograms are performed using the mass spectra and the retention times but not the name provided by the NIST library, which is only informative. Following the BC sample analysis, the chromatograms were scanned for all 761 compounds, without excluding any of them. We aimed to preserve as much information as possible, to maximize the potential insights that could be gained from the data. 

Therefore, for each 2D-chromatogram (Figure 1A), 761 VOCs, further referred to as descriptors, were selected from the corresponding measurement, and the area of their respective peaks was calculated. Each measurement consists of a vector of 761 values and each element of the vector corresponds to the area of a single descriptor (Figure 1B). Each sample was analyzed with randomization and blinding was applied. For each COV sampling on the breast and hands, a maximum of 4 technical replicates were analyzed. 

#### 2.4.2. Statistical Analysis

Statistical analysis was performed using Python 3.8.5. A summary of the statistical pipeline can be consulted in Table 2. All GC × GC − MS data were natural log normalized. The multivariate statistical analysis, namely principal component analysis (PCA), was applied to the sweat volatomics profile dataset to provide insight into the groups. Differences in VOCs between groups were tested with the Mann–Witney U test, and a *p*-value < 0.05 indicated statistical significance without further correction. 

Machine learning (ML) models are employed in volatomic studies trying to identify biomarkers that discriminate between a case and control group. The choice of the most appropriate ML algorithm is a challenging process, therefore the TPOT (Tree-based Pipeline Optimization Tool) was employed [14]. The TPOT is a tool that optimizes ML pipelines using genetic programming. Given the class imbalance between the pre- and post-surgery samples (with fewer samples post-surgery), oversampling was applied. With oversampling, we oversized the minority class by adding observations. Specifically, the SMOTE (Synthetic Minority Over-sampling Technique) algorithm was employed to generate synthetic data for the minority group (post-surgery, see Appendix A) [15]. The TPOT-identified model performance was compared against grid search-optimized logistic regression. Grid search is a tuning technique that attempts to compute a model’s optimum values of its hyperparameters. 

Most of the ML methods are notoriously black box models and therefore difficult to interpret [16]. Applying post hoc interpretable ML methods allows the interpretation of the previously identified best-performing algorithm. Herein, Shapley Additive exPlanations (SHAP)—an interpretable ML method—was employed to determine important VOCs within the model [17]. 

Next, the relevance of the established important VOCs is completed by employing the variable probe method. Within this method, VOCs are ranked given their relevance within the model, and corresponding risk thresholds were applied whereby we can reject candidate VOCs that surpass an associated risk of 5%. Finally, the Jaccard index was employed for gauging the similarity and diversity of sample sets.

## 3. Results

### 3.1. Vector Analysis of Chemical Compounds of Sweat Volatomic Pattern from GC × GC − MS Analysis

Sweat from the hand and breast area of BC patients was analyzed with GC × GC − MS before and after breast-conserving surgery. An example of a chromatogram of sweat VOCs captured on the breast of a single individual is shown in Figure 1A. For each chromatogram, 761 chemical compounds have been identified and the area under the peak has been calculated. Therefore, each measurement consists of a vector of 761 values, an example of measurement on a single individual can be consulted in Figure 1B. For each hand and breast sampling, four or fewer odor-capturing polymer technical replicates were analyzed. 

On both datasets (breast and hand data), the analysis of the distributions of the chromatographic descriptors displays a very variable pattern from one descriptor to another, with notably very different values and frequencies of appearance. The sum of the areas of the raw data is spread over two orders of magnitude for the hand- and breast-collected VOCs over all individuals. Comparing the breast data with the hand data, the sum of the areas is higher for the breast-collected VOCs (Figure 1C,D). Some descriptors from our database systematically appear while others are never observed. The frequencies of occurrence are well distributed over the [0–1] interval with an over-representation of extreme values. The number of non-zero descriptors ranges from minimal 77 to maximal 242 descriptors depending on the breast measurement (Figure 1E). The number of non-zero descriptors fluctuates from a minimum of 82 to a maximum of 348 descriptors per measurement for the hand measurements (Figure 1F).

### 3.2. Multivariate Statistical Analysis of Sweat Volatomics Profile 

#### 3.2.1. A 2-Dimensional Representation of the Datasets

A step-by-step workflow can be consulted in Table 2. PCA was separately performed on the breast and hand VOC data. In a two-dimensional representation of the datasets (both the breast and hand dataset), a clear difference between the measurements was observed and was found to be due to the replacement of one of the GC columns even though a similar brand and reference column was employed (Appendix A). This makes it inappropriate to employ the dataset as they are due to the bias related to the date of analysis which will greatly alter the performance of the model. This technical parameter forces us to divide the full dataset into two datasets, one dataset wherefrom the samples were analyzed with the first GC column and the second dataset with the second column. This subdivision of our dataset will allow for internal validation of our obtained results from one column to another. Once we subdivided our datasets on this parameter, a clear distinction is observed between the pre- and post-surgical state for all patients for both the VOCs collected on the breast as well as the VOCs collected on the hands for both columns (Figure 2 & Appendix A). The pipeline to construct the PCA consists of a natural log standardization with the application of a standard scaler (where the features are standardized by removing the mean and scaling to unit variance). After, for each descriptor, a Mann–Witney U test was performed comparing the pre- and post-operative VOCs where the descriptors were ranked by decreasing *p*-values. Subsequently, for each patient, the chemical compound vector of 761 descriptors is reduced to a vector of significant (*p* < 0.05) descriptors which will serve as the input for the PCA analysis. For further ease of reading, all results presented within the main manuscript are conducted with column 2, and results obtained with column 1 are put into Appendix A. This choice has been made given most data analysis has been conducted with the second employed GC column (further referred to as column 2). 

#### 3.2.2. TPOT

The next step involved supervised learning techniques to predict the pre- vs post-surgery status of the patient based on the measured VOC pattern. Within this approach, the TPOT was used to combine, through genetic algorithms, different ML regressors, and pre-processing algorithms. The TPOT automatically tests them and uses them if they provide more accurate predictions. To compensate for the class imbalance, a SMOTE rebalancing strategy was employed. The rebalancing only takes place on the learning set. For all four datasets (sweat of hands and breasts for both columns) a model is initially fit on a training data set. Successively, the test data set is used to provide an unbiased evaluation of a final model fit (for sample size after implementation with SMOTE, see Appendix A). The models retained are those with the highest F1-weighted cross-validation (CV) score. Here, a stratified 5-fold CV strategy was employed. The tested models and their parameters can be consulted in Appendix A. The TPOT identified logistic regression as a high-performing model with CV scores ranging between 0.69 and 0.93 on the test sets (Table 3 and Appendix A). Logistic regression has the advantage of being simple to implement, requires little computation time, and linearly separates the data. 

#### 3.2.3. Grid Search

ML on small datasets is difficult given model overfitting is likely and it is, therefore, necessary to control the number of model inputs. For this, a grid search on the logistic regression was conducted. Grid search is a method used in hyperparameter tuning to find the best combination of hyperparameters for the model. For the selected logistic regression model, two hyperparameters are fine-tuned: the number of descriptors (hyperparameter N) and a hyperparameter C. The C parameter is employed to balance the trade-off between fitting the training data well and having a simple model with lower variance (less prone to overfitting). Before starting the grid search, a relevance ranking of the descriptors (parameter N) is performed by employing a Mann–Witney U test (ranking from the lowest to the highest *p*-value). Within the grid search, several logistic regressions are trained on the learning/validation set according to the parameters C and N. The best models are those with the highest cross-validation score. If several models have equal performances, the model with the lowest complexity is chosen (C and N as small as possible) to limit the risk of overfitting. The score used is the F1 weighted score of leave-one-out given the limited dataset. For each of the datasets, the F1-weighted CV score was plotted in a contour plot as a function of the C and N parameters (Figure 3 and Appendix A). 

Thus, for the hand data, 175 descriptors were considered whereas for the breast data, only 80 descriptors were taken into consideration for column 2. In the case of the samples analyzed with column 1, only 20 descriptors were necessary for best logistic regression for the breast data whereas 80 descriptors were needed for best logistic regression on the hand data. 

#### 3.2.4. Shapley Values

SHAP values facilitate the explanation of results obtained from supervised ML algorithms. Shapley values help to understand the contribution of each feature to the predictions made by the model by computing the average marginal contribution of a feature over all possible combinations of the other features. An appealing characteristic of SHAP is that an assessment of interaction can be conducted based on visualization rather than complex numerical derivations. Figure 4A shows the descriptors with the highest average impact on the breast data. The higher this impact, the more this descriptor has, on average, a preponderant role in the determination of the “pre- vs post-operative state” character within the VOCs collected on the breast area. Figure 4B shows that each descriptor can have several values and shows the impact associated with each value. As the logistic regression model is linear, the impact of a descriptor is directly linked to its value. For example, the descriptor ‘ethanol, 2-(2-butoxyethoxy)-,acetate’ pushes the prediction towards “post-operative status” when it is associated with a high value and “pre-operative status” when it is associated with a low value within the breast data. Similar calculations have been conducted for the VOCs collected on the hand area and are likewise represented in Figure 4C,D. SHAP results obtained on VOC samples analyzed with the GC column 1 can be consulted in Appendix A. Mainly different descriptors have been identified as important in column 1. The distribution impact of the descriptors on the SHAP outcome for the breast data and hand data with the respective columns can be consulted in Appendix A. One can observe that many descriptors have a low impact on the outcome. 

#### 3.2.5. Probe Variable Method

To determine the relevance of the descriptors, those who predict better than chance, the probe variable method was employed. This method can be summarized as follows: first, a random variable (the probe variable) that does not influence the output is added to the input of the model. After, the SHAP analysis is redone with this new variable. The input variables (descriptors and probe variable) are ranked according to their average impact on the output and the rank of the probe variable is deduced. The lower the rank, the higher the average impact of the probe variable. This method is repeated a large number of times, with the rank of the probe variable depending on the realization. Following these repetitions, the cumulative probability distribution of the rank is obtained. It is thus possible to set a threshold δ that corresponds to the risk of wrongly accepting a descriptor that is less well-ranked than the probe variable, which will consequently be considered irrelevant.

Figure 5A shows the distribution of the rank of this variable after 100,000 realizations on the breast data: the y-axis corresponds to the probability of the probe variable having a rank lower than the one on the x-axis. Three different rejection thresholds are shown for δ = 5% (blue), δ = 10% (red), and δ = 20% (black). For the breast data, only the 80 pre-selected descriptors are considered and the minimum rank of the probe variable is 4 and its maximum rank is 81. With a risk δ of [5–10–20]%, [27–28–33] descriptors are kept, respectively. In the case of the hand dataset (Figure 5B), the minimum rank of the probe variable is 2 and its maximum rank is 176. With a risk δ of 5%, 10%, 20%, 16, 25, and 35 descriptors are kept, respectively. For the breast data analyzed with column 1, a risk δ of [5–10–20]%, [7–8–9] descriptors are kept, respectively, with a minimum and maximum rank of 1 and 21. For the hand data analyzed with column 1, a risk δ of 5%, 10%, 20%, 21, 28, and 35 descriptors are kept with a minimum and maximum rank of 1 and 81, correspondingly (Appendix A). 

Finally, by combining SHAP with the probe variable method, about 20–40 descriptors relevant for predicting pre- and postoperative character were identified on the hand data, and 20–30 descriptors were relevant on VOCs captured on the breast. To find out if the same descriptors are retained for both datasets, the Jaccard index of retained descriptors (length of intersection/length of the union) against the number of retained descriptors was plotted. A high index indicates that the descriptors are globally the same and a low index identifies that retained descriptors are different. A small number of descriptors are in common between the VOCs identified as relevant for the pre-post-surgery distinction for the hand dataset vs. the relevant ones identified on the breast dataset (Figure 6). A similar comparison was made between similar body parts across different columns (Appendix A). For both the hand VOC dataset as well as the breast VOC dataset across different columns, a small but certain number of VOCs were identified as commonly relevant to differentiate pre- and post-operative conditions (Appendix A). Individual compound names are available in Appendix A, along with their feature rank and feature importance value based on SHAP. In addition, the fraction of occurrence for both the ‘sick’ and ‘healthy’ status and the maximum absolute difference between both states are provided. Individual tables contain information for either the hand or breast dataset analyzed with their respective GC column.

## 4. Discussion

BC is a global problem and better detection tools for screening are required. The analysis of VOCs provides an alternative that could complement the current BC screening tools. However, to date, there is no uniform standard for candidate BC biomarkers in expelled VOCs from BC patients [4]. We have analyzed the VOCs in sweat on the hand and breast area of BC patients before and after tumor-removing surgery employing GC × GC − MS technology to obtain comprehensive volatile fingerprints of BC. This experimental setup and analysis allowed the detection of expelled VOCs that are potentially linked to the BC tumor presence. In addition, the analysis of the sweat of BC patients before and after surgery from the same subject aids in balancing the effect of external interferents, such as diet or environmental exposure.

### 4.1. Pipeline for Identification of BC-Related VOCs

For the discovery of biomarkers in complex chemical data sets, commonly multivariate methods such as partial least-squares to latent structures (PLS), orthogonal PLS, or PCA are employed [4]. Within this research, we tried pushing further to optimize the pipeline by selecting the highest-performing ML methods utilizing TPOT. Additionally, increased interpretability of the selected model was obtained with the use of SHAP and the variable probe method. 

Considering that the dataset contains a limited number of measures and patients, a methodology of strict control of the risk of overlearning was implemented. Satisfactory classification performances were obtained with a logistic regression: the F1-weighted scores on the test set are at 93% and 85% for the breast dataset and the hand dataset in column 2, respectively. Lower F1 scores were obtained for column 1, likely due to the smaller data availability to construct the ML models. Importantly, high sensitivity rates (minimum 96%) were obtained for all the datasets across the different columns. For both columns, a lower number of descriptors was identified as important within the breast datasets to obtain high F1 scores. Here, it should be noted that the VOC database was constructed on VOCs collected on the hands and therefore there might be a bias in the preselection of VOCs that have been optimized to favor the hand dataset. It might be seen as a positive confirmation that a database developed on hand VOCs identifies more important VOCs within the hand area compared with those of other regions. It is likely that for the same reason, more compounds are identified as relevant (variable probe method) for both columns on the hand area and fewer on the breast area. It should therefore be noted that VOCs which might be relevant for BC in breast sweat might be missing from a dataset built mostly on hands. In the future, it will be necessary to complement the current database with an in-depth analysis of VOCs collected specifically on the breast area. 

### 4.2. VOCs Matrices

Afterward, the SHAP method coupled with the probe variable method allowed the selection of important and relevant descriptors for the VOCs collected on the hands and breasts. However, these descriptors’ records have little in common, suggesting that the body parts used for sampling have a significant impact on the VOCs distinguishing the pre- and post-surgery status (Appendix A). Furthermore, a small number of VOCs were found to have a large impact following the SHAP analysis on the outcome for both the hand and breast models (Appendix A). VOCs from the skin are derived from sebaceous, eccrine, and apocrine gland secretions. These glands are differently dispersed across the body; hence different regions of the body have distinct VOC profiles, and thus diverse odors. Eccrine glands are concentrated in the palms of hands and produce sweat that mostly consists of water and glycoproteins. Whereas the upper chest consists of many sebaceous glands (absent on the palms of hands) [18] whose secretions are rich in lipid materials [19]. The currently employed VOC database was constructed on VOCs collected in the hand area. To collect more VOCs in the breast area, a VOC database on the breast area should be constructed. Furthermore, multiple studies have demonstrated that sweat VOC composition alters with age [20], making it important to correctly balance age classes between healthy subjects and patients in clinical volatomic studies. Within this study, sweat was chosen, given its close physical proximity to the breast tumor. However, other matrices could be of interest. Systematic research on BC VOCs identified sixteen GC − MS BC case-control studies, from which four studies analyzed VOCs in urine [4]. Urine has been recognized as a valuable matrix for volatomic BC diagnosis. This is because most substances are metabolized in the liver and excreted in the urine, allowing it to contain important information about the clinical condition of the organism. 

### 4.3. Confounding Parameters

BC is a heterogeneous disease. Molecular subgrouping can be provided by using several immunohistochemical markers [21]. It is reasonable to assume that the VOC profile would be different depending on the molecular subtypes of BC, given the metabolic expression of distinct markers to sustain fast cell growth and proliferation [22,23]. Within this study, there was no attempt to differentiate among different BCs. The presence of different molecular subtypes and clinical grades may affect the experimental results. In future studies, we will expand our sample size and differentiate among pathological types and stages of tumors, allowing for more accurate volatile biomarkers to be identified. Most available studies to date are cross-sectional and case-control studies. However, ideally, cancer-specific biomarkers need to be investigated in prospective longitudinal studies to understand to what extent the VOCs are associated with the disease severity. 

Additionally, VOCs reflect metabolic factors, so interference should be precisely controlled, such as medication, diet, and exercise. Within our study, even if separation is observed between the pre- and post-surgery statutes, the results should be taken with caution. Differentiation is possibly caused by the tumor removal, but may also be related to environmental differences that we cannot control. Therefore, more relevant tests are required to establish the standards and conditions for possibly different confounding factors. One suggestion would be to collect multiple samples for each condition, allowing to better distinguish what might be the most relevant BC VOCs. Within our experiment, we had the opportunity to test the same experimental setup over two different GC columns, allowing for internal verification of the obtained results. As demonstrated, only a core set of VOCs (Appendix A) has been similarly identified over the two different experimental settings, highlighting the high need for multi-centered analysis of VOCs in future volatomics studies. 

### 4.4. Instrumental Parameters

For each chromatogram, 761 chemical compounds were selected. Importantly, the vector is a post-processing of 2D chromatograms with loss of information and presence of *a priori* of the scientist on the chemical compounds of potential interest. Though thermal desorption coupled with comprehensive two-dimensional gas chromatography is satisfactory from a separative point of view, low-resolution MS does not allow the detected molecules to be identified with high certainty. The chemical identities of the compounds shown should therefore be regarded as tentative, given their structures were inferred from resemblances between their mass spectra in the computer-based NIST library (with minimum resemblances percentages of 80%). Although widely employed as an analytical tool, this method is prone to error. Within this research, we aimed to exhibit state-of-the-art BC sweat volatomics with a unique pre-post-surgical experimental setting, processed with an innovative analysis pipeline. Therefore, little importance has been put into the identification of the VOCs and we do not further exploit the selection of identified VOCs distinguishing between pre-post-surgery status. In the future, we aim to confirm the chemical identities of candidate VOC BC biomarkers with high-resolution devices (thermal desorption coupled with GC×GC-high-resolution MS), allowing for a more formal identification that is necessary for medical diagnostic purposes. 

Nonetheless, GC − MS technology is considered the gold standard within the volatomics field. However, there are limitations to this technique: the instrument is expensive, and the analysis is time-consuming. E-noses have the potential to overcome these disadvantages, given that they provide a cheap, fast, and portable way to analyze gas samples. Additionally, detection dogs can potentially aid in pathology detection based on specific VOCs [24]. However, the sensibility of different animal species to chemical substances has been shown to be heterogeneous [25]. Today, there is a lack of biomarker identification for volatomics analysis, making high-performance equipment indispensable to expand our basic knowledge. Once a disease-associated volatile signature is identified, it is more likely that the forthcoming approach to be used in clinical settings might be the E-nose or animal detection method, due to their ease and cost of testing over GC − MS. 

## 5. Conclusions

This study tackled the untargeted assessment of volatomics sweat profile from BC patients using GC × GC − MS combined with an innovative multivariate statistical toolkit (PCA, TPOT, SHAP, probe variable method). A total of 761 volatiles were identified and statistical analysis revealed that metabolites were altered in the sweat of BC patients in their pre- vs post-surgery status. Sweat VOC BC testing may prove useful as an addition to population screening with mammography by identifying a low-risk group that can be offered less intensive screening or could be effectively used in disease screening programs in clinical setups across developing nations. 

The VOC identification within this research should be regarded as tentative given the low patient number, potentially non-excluded confounding factors, and the analysis with low-resolution MS. In the future, we aim to conduct validation studies in larger and more diversified populations with a multi-centered analysis setting employing high-resolution devices to confirm the obtained findings. Nonetheless, the analysis gives encouraging insights into the potential of VOC sweat analysis for BC screening. These results strongly support the idea that endogenous metabolites can be used as potential BC biomarkers, opening the way to new screening tools.

## Figures and Tables

**Figure 1 cancers-15-02939-f001:**
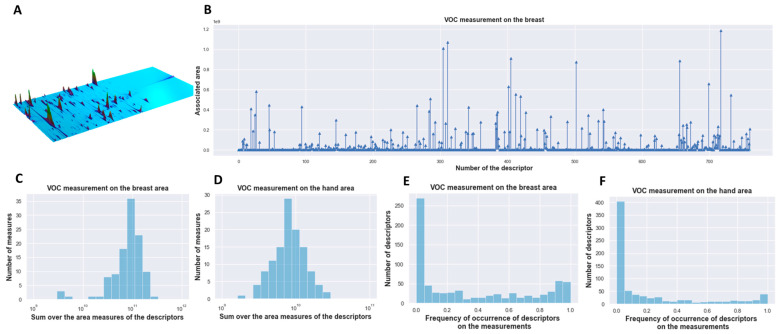
Overview & descriptive analysis of GC × GC − MS data. (**A**) An example of a chromatogram of sweat VOCs captured on the breast of a single individual. (**B**) Example of chromatogram on VOCs captured on the breast area and converted into a vectorized measurement of a single individual. (**C**) Sum over the area measures of the descriptors on the breast and (**D**) hand data over all individuals. (**E**) Frequency occurrence of descriptors on the breast and (**F**) hand area measurement over all individuals.

**Figure 2 cancers-15-02939-f002:**
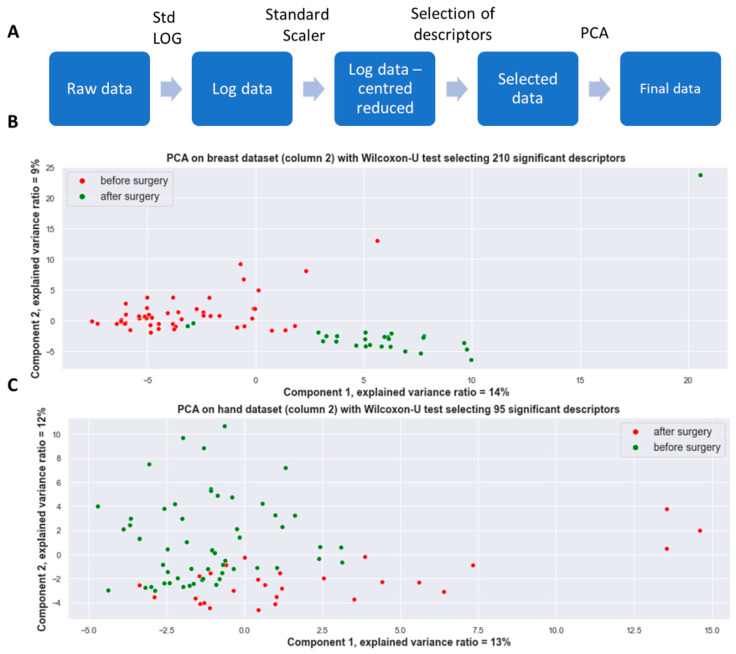
Principal component analysis of VOCs collected on breast and hand area (column 2). (**A**) Employed pipeline for PCA construction. (**B**) PCA on the breast dataset with Mann–Witney U descriptor selection for column 2. (**C**) PCA on the hand dataset with Mann–Witney U descriptor selection for column 2.

**Figure 3 cancers-15-02939-f003:**
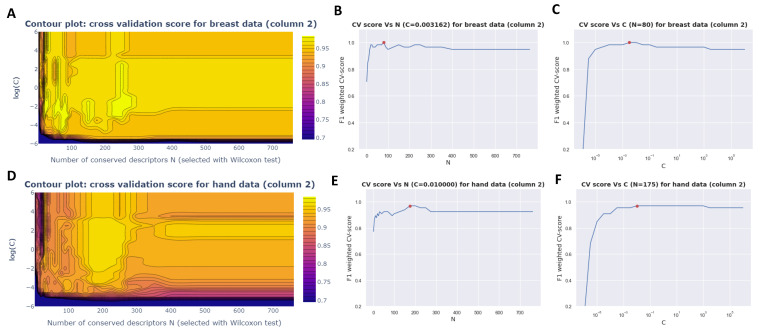
Grid search on logistic regression on breast and hand VOC dataset (column 2). (**A**) shows the contour plot for the breast dataset. (**B**,**C**). The best model with the lowest complexity is obtained for C = 0.003162277 (Figure 3B) and N = 80 (Figure 3C) where a CV-score of 1 is obtained. (**D**) The contour plot for the hand dataset (**E**,**F**). For the VOCs collected on the hands, the best model with the lowest complexity is obtained with C = 0.01 (Figure 3E) and N = 175 (Figure 3F).

**Figure 4 cancers-15-02939-f004:**
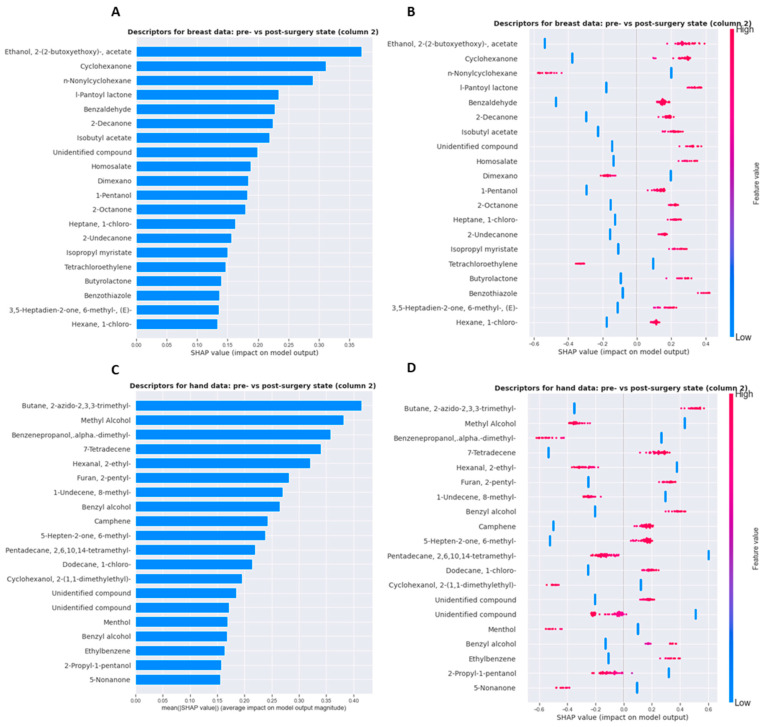
SHAP analysis. SHAP summary plot for the (**A**) breast data (GC column 2) and (**C**) hand data (GC column 2). (**B**) Detailed summary plot (SHAP value: negative = “pre-surgery”, positive = ”post-surgery”) for the VOCs collected on the breast area and (**D**) hand area. High feature values are visible in red while low feature values are displayed in blue.

**Figure 5 cancers-15-02939-f005:**
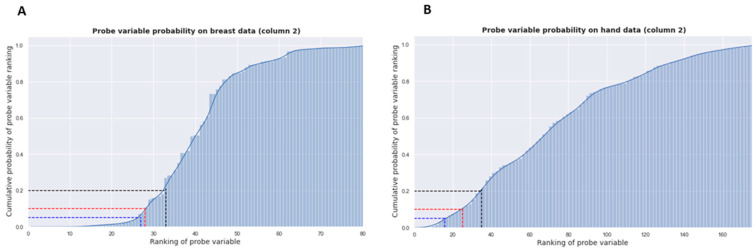
Probe variable method for (**A**) the breast data and (**B**) the hand dataset on VOC analysis conducted with GC column 2.

**Figure 6 cancers-15-02939-f006:**
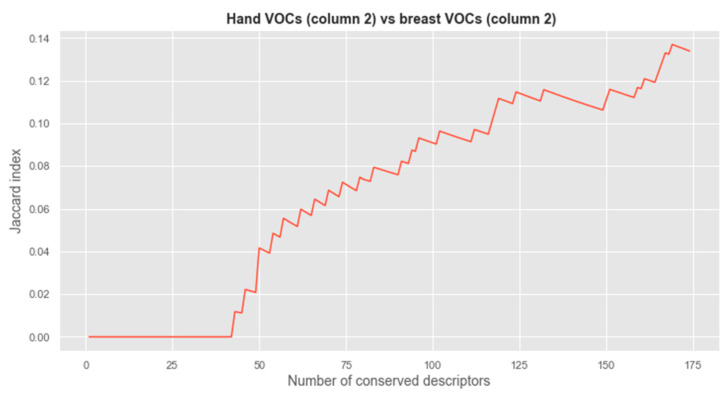
Jaccard index of the retained descriptors between hand-collected VOCs and breast-collected VOCs with column 2.

**Table 1 cancers-15-02939-t001:** Collected sweat samples from BC patients on the hand and breast area before and after surgery. For each sampling, 4 or fewer odor-capturing polymers were taken as technical replicates and analyzed.

	Sweat Samples before Surgery		Sweat Samples after Surgery	
Body part sweat collection	Hands	Breast	Hands	Breast
Subjects	21	20	13	12
Number of samples	74	70	43	41
Age (range, median)	mean 51, range (36–76)	mean 51, range (36–76)	mean 52, range (41–76)	mean 51, range (36–71)

**Table 2 cancers-15-02939-t002:** Workflow data analysis VOCs.

1.	Application of LOG chemical standardization.
2.	Principal component analysis: visualization of clusters of samples based on their similarity.
3.	Split the data into two sets: a learning/validation set and a test set and rebalance the classes/labels on the learning set employing SMOTE.
4.	Train models with TPOT.
5.	Evaluation of the performance of the selected models by computing their F1-weighted score on the test set
6.	Interpret the model using SHAP
7.	Determine the relevance of VOCs using the probe variable method

**Table 3 cancers-15-02939-t003:** Evaluation of logistic regression performance on the breast and hand dataset employing GC column 2.

	Pre-vs. Post-Surgery Status on Breast VOCs (GC Column 2)	Pre-vs. Post-Surgery Status on Hand VOCs (GC Column 2)
F1-score	0.93	0.85
Sensitivity	1.0	0.96
Specificity	0.82	0.64

## Data Availability

The data presented in this study are available on request from the corresponding author. The data are not publicly available due to privacy restrictions.

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
