# Peer review of "Screening of Breast Cancer from Sweat Samples Analyzed by 2-Dimensional Gas Chromatography-Mass Spectrometry: A Preliminary Study"

_cancers, 2023, doi:10.3390/cancers15112939_

Round 1
Reviewer 1 Report
The authors proposed using a wearable medium for the non-invasive sampling followed by the VOC profiling by the GC×GC-MS system, which is a promising method for comprehensively capturing the volatile features associated with breast cancer progression or prognosis. However, there are still some serious issues that remain unfixed from the methodology point. A major revision is suggested before considering the acceptance. Here are some comments that need the authors’ addressing.
1. The quality control processes for the sample/data result are lacking.
2. Will the wearable sampling medium also generate some detectable VOC peaks? If so, these false positive peaks will be mixed into the descriptors and interfere with the follow-up data processing and final classification. Have the authors ever investigated this issue? Is there any way to subtract the background peaks from the raw 2D MS spectrum? It is suggested to acknowledge this issue and the corresponding solution in the discussion or method description.
3. Line 203-204, is there any specific criterion that excludes a peak when its occurrence frequency is lower than a certain threshold, such as 0.1, 0.2, 0.5, etc? Please specify it.
4. Line 205-208, the non-zero descriptors in the hand/breast measurement have a very wide range from 77/82 to 242/348. It was also stated that there are 761 descriptors in each measurement presenting the VOC profile (Line 195, Line 229). It is confusing how many common peaks are used exactly to present the VOC profile. What is the criterion to define a common peak that exists in all or most samples (percentage, or the occurrence frequency that the authors mentioned in Line 203-204).
5. The method section describes the peak identification by NIST 14 library with a matching score above 80%. But in the abstract, it was stated that “761 volatiles from a home-made human odor library were screened on each chromatogram.” (line 33-34). These statements seem contradictory with each other.
6. The sweat sampling amount may have influence on the VOC peak intensities. How did the authors handle the sampling variation? Is there any internal standard or normalization method employed?
7. In supplementary Tables 6 and 7, the contents of identified VOCs are too brief to make readers learn something meaningful. It is suggested to add at least their importance (say, weight coefficient in the classification model) in the machine learning model. So, the reader may learn which compound deserves special attention or further study.
8. Some typos need to be corrected. “COV identification” (Line 500, VOC?); “TD-GC×GC-MS” (Line 466 and 478, 2D-GC×GC-MS?)
Generally looks good with just a few typos.
Reviewer 2 Report
The manuscript from Leemans et al. describes a proof-of-concept study investigating volatile organic compounds (VOCs) emitted through the metabolism of breast cancer cells as possible novel cancer biomarkers for non-invasive breast cancer screening. The study is well documented, and the innovative multivariate statistical toolkit is very well appreciated. However, as the authors stated, a much larger sample cohort with more heterogeneous people, comprising a validation cohort of cancer and non-cancer samples, is needed to confirm the results obtained in this study.
See my comments below:
- Page 2, lines 60-61 “Volatilomics emerged as a non-invasive approach, potentially low-cost, with a potential larger coverage of the population for the cancer screening.” Please include here references.
- Page 2, lines 67- 68 “Gas Chromatography coupled with Mass Spectrometry (GC-MS) is the gold standard for identifying and quantifying human-expelled VOCs.” Please include a reference.
- Introduction. Page 2 line 75 -76 “This study aimed to establish the volatomic signature of sweat in BC. To this end, sweat of BC patients before and after tumor-removing breast surgery was analyzed with 2D-GC-MS technology combined with multivariate statistical tools„. What is the rationale here? Why the authors want to compare the sweat of BC patients before and after tumor-removing breast surgery? Do the authors refer the sweat after tumor-removing breast surgery as a non-tumor sweat sample? What do the authors expect from this comparison? More information is needed here.
- Subsection 1.1.: In the inclusion criteria, did the patients take anti-cancer drugs before surgery when the sweats were collected? Also, were the breast cancers recurred? Please include these information in the manuscript.
- Table1: what the “Number of samples” stays for? Is the sweat sample for each subject divided in more samples? Please, clarify here and in the manuscript.
- Page 3 line 132: which devise was used for thermodesorption?
Minor:
- Page 2, line 72: “… allowing for additional separation of COVs that might coelute…” please define “COVs”, maybe the authors meant “VOC”? If so, please correct also elsewhere in the text.
- Page 4 line 174 “… see Table 1 and Supplementary 2… is this a table? If so, it should be “Supplementary Table 1” please correct.
- Figure 3: There is a discrepancy between the Figure 3B, 3C showed in the picture and the Figure 3B, 3C described in the caption. In the caption it is stated “The best model with the lowest complexity is obtained for N=80 (Figure 3B) and C=0.003162277 (Figure 3C)” but in the picture it is shown the opposite. Same for Figure 3E and 3F. Please correct and clarify.
- Page 6 lines 269-270: it is stated “In the case of the samples analyzed with column 1, only 10 descriptors were necessary for best logistic regression”, however, Supplementary Figure 3 showed 20 descriptors for the breast data. Please clarify.
Round 2
Reviewer 1 Report
The authors have well addressed all the reviewer's comments and concerns, which is appreciated. I agree that the manuscript can be accepted for publication.